# A Novel Ultra−High Resolution Imaging Algorithm Based on the Accurate High−Order 2−D Spectrum for Space−Borne SAR

**Tao He [1], Lei Cui [2], Pengbo Wang [1,*], Yanan Guo [1] and Lei Zhuang [2]**

[1] School of Electronics and Information Engineering, Beihang University, Beijing 100191, China; tao_h@buaa.edu.cn (T.H.); gyanan@buaa.edu.cn (Y.G.)
[2] Shanghai Institute of Satellite Engineering, Shanghai 201109, China; clei117@163.com (L.C.); zhuanglei0411@163.com (L.Z.)
[*] Correspondence: wangpb7966@buaa.edu.cn; Tel.: +86-10-823-386-70

**Abstract:** Ultra−high spatial resolution, which can bring more detail to ground observation, is a constant pursuit of the modern space−borne synthetic aperture radar. However, the exact imaging in this case has always been a complex technical problem due to its complicated imaging geometry and signal structure. To achieve those applications' strict requirements, a novel ultra−high resolution imaging algorithm based on an accurate high−order 2−D spectrum is presented in this paper. The only first two Doppler parameters needed as range models in the defective spectrum are replaced by a polynomial range model, which can derive coefficients from the relative motion between the radar and the targets. Then, the new spectrum is calculated through the Lagrange inversion formula. Based on this, the novel imaging algorithm is elaborated in detail as follows: The range high−order term of the spectrum is compensated completely, and the range chirp rate space variance is eliminated by the cubic phase term. Two steps of range cell migration correct are applied in this algorithm before and after the range compression; one is the traditional linear chirp scaling method, and another is the interpolation to correct the quadratic range cell migration introduced by the range chirp rate equalization. The simulation results illustrate that the proposed algorithm can handle the exact imaging processing with a 0.25 m resolution around the azimuth and range in 2 km × 6 km, which validates the feasibility of the proposed algorithm.

**Keywords:** space−borne SAR; ultra−high resolution; high−order 2−D spectrum; Lagrange inversion formula; chirp rate space variance

## 1. Introduction

Space−borne synthetic aperture radar (SAR) is an ideal approach in earth observation, which has risen in recent years, both in military and civil applications, due to its advantages of capability with all−time and all−weather. To receive more detailed information in SAR images, the requirement of the resolution becomes higher. Currently, many advanced orbiting space−borne SARs such as TerraSAR−X [1–3], Cosmo−SkyMed [4,5], and GaoFen−3 are capable of reaching the 1 m resolution, and the TerraSAR NEXT GENERATION (TerraSAR NG) [6] plans to enhance this ability to 0.25 m.

In the traditional space−borne SAR system, many classic imaging algorithms possess great ability with the simple air−borne imaging geometry assumptions like the range Doppler algorithm (RDA) [7] and the chirp scaling algorithm (CSA) [8]. The hyperbolic range equation model (HREM) [9] or equivalent squint range model (ESRM) [10] applied in RDA and CSA only require the Doppler centroid and Doppler rate, which supposes the satellite trajectory as a straight line during the integration time. Meanwhile, these methods regard the three− and higher−terms of the signal spectrum as the phase error and ignore them. Conversely, the wavenumber domain algorithm (ωkA) [11] is a classic

high−resolution imaging processor. There is no range cell migration (RCM) approximation in ωkA, and the Stolt interpolation is used to finish the range cell migration correction (RCMC). However, along with the increasing resolution, the apparent curve of the satellite trajectory appears in the long duration time, the high−order phase terms are no longer negligible, and the Stolt interpolation will be highly cumbersome in the immense volume of data. Furthermore, back−projection algorithm (BPA) [12] is another accurate image processor applied in the time domain; it utilizes every radar pulse back−projected in every pixel in the imaging scene to acquire the SAR image exactly. However, the classic BPA requires $N \times N \times N$ times back−projection operations, while $N$ is the number of the azimuth pulses (suppose the azimuth samples are equal to the range samples). This is a large burden for the computers. Many researchers have optimized it and proposed relatively efficient methods such as the fast factorized back−projection algorithm (FFBPA) [13] and the modified FFBPA [14,15], but they still have a considerable amount of calculation.

Under the requirements of processing efficiency and accuracy, many scholars have focused their research on the improvement of the frequency domain imaging algorithm. A four−order processing algorithm was proposed by Eldhuset [16–18] in 1998, which established an extended exact transfer function by the four−order range model in the space−borne SAR geometry. It assumes the RCM does not vary in the one data block so that the RCMC and the range compression are completed together in the 2−D frequency domain and achieve the targets well−focusing with the resolution around 1 m after the azimuth compression. To obtain further results, in 2013, Wang [19] applied this Eldhuset's range model in the modified chirp scaling algorithm. The new four−order 2−D spectrum is derived and improves the imaging algorithm to a large extent, which finally achieved results under 0.3 m. This is a tremendous improvement from the conventional CSA, but its performance is still limited because the spectrum term more than five−order is neglected. Luo [20] proposed an extended range Doppler domain algorithm in 2013. A sinc interpolation is applied in this method to remove the differential RCM and uses the range segmenting to strengthen the focus depth by eliminating the residual cross−coupling phase increased with the slant range displacement. This algorithm has a large image width (above 10 km) in range direction and reaches the resolution around 0.3 m. A year later, another advanced imaging algorithm evolved from the hybrid correlation algorithm was proposed by Wang [21]. A modified ESRM is proposed in this paper by adding an equivalent radar acceleration, and the hybrid correlation factor is constructed through it. It starts with the coarse focusing by the azimuth reference function multiplication to remove the bulk RCM, and the differential RCM is then corrected through the hybrid correlation processing in the refined focusing. A higher resolution is obtained in this method, which is near 0.25 m.

It is obvious that a significant improvement is made in the modified CSA [19], but its performance is limited along the azimuth resolution increasing due to the insufficient 2−D spectrum. The last two methods make a breakthrough at the resolution, but they all avoid the 2−D spectrum calculation and apply their kernels in the range time domain most of the time. However, the accurate 2−D spectrum is necessary to research the ultra−high−resolution space−borne SAR. It contains much important information about the echo properties. This paper proposes an accurate high−order 2−D spectrum brought by the Lagrange inversion formula to meet the above requirements. The ordinary HREM or ESRM is replaced by an infinite polynomial range model, while coefficients can be computed by the satellite and target's relative motion vectors, or obtained from other high−precision range models through the Taylor series directly. After this, the principle of stationary phase (POSP) [22] is applied to both the range and azimuth directions, and the Lagrange inversion, which is widely used in the complex spectrum calculation for SAR imaging [23,24], is employed to obtain the analytical expression of the spectrum.

Based on the aforementioned accurate 2−D spectrum, a novel ultra−high−resolution imaging algorithm is presented as follows. The range high−order term is derived by the Taylor series on the proposed spectrum, directly and completely compensated. Moreover, drawing lessons from the principle of chirp scaling [8], the RCM caused by the relative mo-

tion between the satellite and the targets is accomplished by complex multiplication in the RD domain. Furthermore, to get enough focus depth, the range chirp rate's space variance, which will render the range defocused, is non−negligible in this case. A non−linear FM filter method proposed by Davidson [25] is a common approach to compensate the residual phase raised by the squint angle in the quadratic range compression (SRC). It was validated through the simulation based on Seasat and ERS−1 and achieves the well−focusing with squint angle of up to 35 deg for L−band and 50 deg for C−band. Moreover, the article mentions a limitation: the reference Doppler frequency must be selected outside of the Doppler bandwidth, which will introduce the extra phase error. Another efficient method proposed by Wong [26] in 2008 is applied in this paper. The range chirp rate is assumed to vary linearly, so the cubic phase function can equalize the dominant quadratic phase term of each target, which is like the chirp scaling principle. Nevertheless, there is still a problem that needs to be handled. The residual linear phase introduced above would cause the chirp phase center to shift in range direction with the Doppler frequency and be reflected as the quadratic RCM after the range compression due to the corresponding square of range difference. This quadratic RCM is usually neglected because of the slight variation of chirp rate for low resolution. In contrast, it must be corrected for our research case, so the interpolation is applied to fulfill the correction and finish the range processing. Finally, the well−focused targets can be obtained after the azimuth processing.

This paper is organized as follows: The Infinite polynomial range model and the accurate high−order signal spectrum are introduced in Section 2. In Section 3, the proposed imaging algorithm for the ultra−high space−borne SAR is described in detail. In addition, an analysis of the proposed algorithm's performance and an integrated simulation is presented in Section 4. Finally, conclusions are drawn in Section 5.

## 2. Infinite Polynomial Range Model and Signal Spectrum

Eldhuset gives a four−order range model to describe the SAR range history in [16], but it only corresponds to the first four−order relative motion parameters between the satellite and the targets. To describe the exact range history of the long illumination time, a general infinite polynomial is presented in this paper. The range history vector $\overrightarrow{R}(\eta)$ [16] can be expressed through the Taylor expansion in azimuth time $\eta$ as

$$\overrightarrow{R}(\eta) = \sum_{i=0}^{\infty} \frac{1}{i!} \overrightarrow{X}_i \eta^i \tag{1}$$

while $\overrightarrow{X}_i$ is the $i$th−order relative motion parameters. Modulo both sides of Equation (1) and processed through the binomial theorem, the scalar value of the range history can be acquired in Equation (2).

$$R(\eta) = \sqrt{\left|\overrightarrow{R}(\eta)\right|^2} = \sqrt{\left(\sum_{i=0}^{\infty} \frac{1}{i!} \overrightarrow{X}_i \eta^i\right) \cdot \left(\sum_{i=0}^{\infty} \frac{1}{i!} \overrightarrow{X}_i \eta^i\right)} = \sqrt{\sum_{i=0}^{\infty} Y_i \eta^i} \tag{2}$$

while

$$\begin{cases} Y_0 = \overrightarrow{X}_0 \cdot \overrightarrow{X}_0 \\ Y_1 = 2\overrightarrow{X}_0 \cdot \overrightarrow{X}_1 \\ Y_2 = \overrightarrow{X}_1 \cdot \overrightarrow{X}_1 + \overrightarrow{X}_0 \cdot \overrightarrow{X}_2 \\ Y_3 = \frac{1}{3} \overrightarrow{X}_0 \cdot \overrightarrow{X}_3 + \overrightarrow{X}_1 \cdot \overrightarrow{X}_2 \\ Y_4 = \frac{1}{4} \overrightarrow{X}_2 \cdot \overrightarrow{X}_2 + \frac{1}{12} \overrightarrow{X}_0 \cdot \overrightarrow{X}_4 + \frac{1}{3} \overrightarrow{X}_1 \cdot \overrightarrow{X}_3 \\ Y_i = \sum_{n=0}^{i} \left( \frac{1}{n!} \overrightarrow{X}_n \frac{1}{(i-n)!} \overrightarrow{X}_{i-n} \right) \end{cases} \tag{3}$$

Expanding Equation (2) by the Taylor series, the no error polynomial range model is given as

$$R(\eta) = \sum_{i=0}^{\infty} R_i \eta^i, \text{ where } R_i = \frac{R^{(i)}(0)}{i!} \tag{4}$$

while $R_i$ represents the $i$th$-$order range model coefficient.

The space$-$borne SAR received echo from a point target, after demodulated to the baseband, can be derived as

$$s(\tau, \eta) = \sigma_0 \omega_r \left( \tau - \frac{2R(\eta)}{c} \right) \omega_a(\eta - \eta_c) exp \left\{ -j\frac{4\pi f_0 R(\eta)}{c} \right\} exp \left\{ j\pi K_r \left( \tau - \frac{2R(\eta)}{c} \right)^2 \right\} \tag{5}$$

where the $\sigma_0$ represents the target's back scattering coefficient; $\omega_r(\cdot)$ and $\omega_a(\cdot)$ denote the antenna pattern functions on the range and azimuth directions, respectively; $c$ is the velocity of light; $\eta_c$ is the Doppler central time; $f_0$ is the signal carrier frequency; $K_r$ is the range chirp rate; $\tau$ is the range time.

To acquire the 2$-$D spectrum of the echo, the POSP is applied. It is easy to achieve this operation on the range direction and obtain the signal in the range$-$frequency (RD) domain as

$$S(f_\tau, \eta) = \sigma_0 \omega_r(f_\tau) \omega_a(\eta - \eta_c) exp \left\{ -j\frac{4\pi(f_0 + f_\tau)R(\eta)}{c} \right\} exp \left\{ -j\frac{\pi f_\tau^2}{K_r} \right\} \tag{6}$$

where the $f_\tau$ is the range frequency. However, in the azimuth direction, the stationary phase point cannot work out through differentiating $\eta$ on the integrand phase Equation (7) directly, so we must find other ways to solve this problem.

$$\theta(\eta) = -\frac{4\pi(f_0 + f_\tau)R(\eta)}{c} - 2\pi f_\eta \eta = -\frac{4\pi(f_0 + f_\tau)}{c} \sum_{i=0}^{\infty} R_i \eta^i - 2\pi f_\eta \eta \tag{7}$$

According to the time$-$frequency properties of the Fourier transform, the linear term multiplied in the time domain is equivalent to a frequency shift. Therefore, we can extract the linear term in Equation (3) and acquire the new form of the signal as

$$S(f_\tau, \eta) = S'(f_\tau, \eta) exp \left\{ -j\frac{4\pi(f_0 + f_\tau)R_1\eta}{c} \right\} \tag{8}$$

while

$$S'(f_\tau, \eta) = \sigma_0 \omega_r(f_\tau) \omega_a(\eta - \eta_c) exp \left\{ -j\frac{4\pi(f_0 + f_\tau)}{c} \sum_{\substack{i=0 \\ i \neq 1}}^{\infty} R_i \eta^i \right\} exp \left\{ -j\frac{\pi f_\tau^2}{K_r} \right\} \tag{9}$$

The new integrand phase of Equation (9) can be expressed as Equation (10), and the stationary phase point can be derived from its differential.

$$\theta(\eta) = -\frac{4\pi(f_0 + f_\tau)}{c} \sum_{\substack{i=0 \\ i \neq 1}}^{\infty} R_i \eta^i - 2\pi f_\eta \eta \tag{10}$$

$$\frac{d\theta(\eta)}{d\eta} = -\frac{4\pi(f_0 + f_\tau)}{c} \frac{d}{d\eta} \sum_{\substack{i=0 \\ i \neq 1}}^{\infty} R_i \eta^i - 2\pi f_\eta = -\frac{4\pi(f_0 + f_\tau)}{c} \sum_{i=2}^{\infty} i R_i \eta^{i-1} - 2\pi f_\eta = 0 \tag{11}$$

To make the followed steps more intuitive, variable $P$ is used to simplify the differential equation as

$$P = \sum_{i=2}^{\infty} i R_i \eta^{i-1} \tag{12}$$

while

$$P = -\frac{c f_\eta}{2(f_0 + f_\tau)} \tag{13}$$

Obviously, Equation (12) is an infinite series without any constant term. Therefore, we can use the Lagrange inversion [27] to obtain its inverse function as another infinite series by

$$\eta = \sum_{i=2}^{\infty} C_i P^{i-1} \tag{14}$$

while

$$\begin{cases} C_2 = \frac{1}{2R_2} \\ C_3 = -\frac{3R_3}{8R_2^3} \\ C_4 = \frac{9R_3^2 - 4R_2R_4}{16R_2^5} \\ C_5 = \frac{-20R_5R_2^2 + 120R_4R_2R_3 - 135R_3^3}{128R_2^7} \\ C_6 = \frac{-24R_6R_2^3 + \left(96R_4^2 + 180R_3R_5\right)R_2^2 - 756R_3^2R_4R_2 + 567R_3^4}{256R_2^9} \\ \cdots \end{cases} \tag{15}$$

So that the 2$-$D spectrum of $S'(f_\tau, \eta)$ can be achieved by introducing Equation (14) to Equation (10), and the result is given as

$$S'(f_\tau, f_\eta) = \sigma_0 \omega_r(f_\tau) \omega_a \left( f_\eta - f_{\eta_c} - \frac{2(f_0 + f_\tau)R_1}{c} \right) exp\{ j\Phi'(f_\tau, f_\eta) \} \tag{16}$$

$$\begin{aligned} \Phi'(f_\tau, f_\eta) = & -\frac{4\pi(f_0 + f_\tau)}{c} \sum_{i=2}^{+\infty} \frac{i-1}{i} C_i \left[ P(f_\eta) \right]^i \\ & -2\pi f_\eta \sum_{i=2}^{+\infty} C_i \left[ P(f_\eta) \right]^{i-1} - \frac{\pi f_\tau^2}{K_r} - \frac{4\pi(f_0 + f_\tau)R_0}{c} \end{aligned} \tag{17}$$

Finally, the linear term $R_1 \eta$ is added as a frequency bias into the Equation (16) and the precise high$-$order 2$-$D spectrum of $s(\tau, \eta)$ is elaborated as

$$S(f_\tau, f_\eta) = \sigma_0 \omega_r(f_\tau) \omega_a(f_\eta - f_{\eta_c}) exp\{ j\Phi(f_\tau, f_\eta) \} \tag{18}$$

while

$$\begin{aligned} \Phi(f_\tau, f_\eta) &= \Phi'\left( f_\tau, f_\eta + \frac{2\pi(f_0 + f_\tau)R_1}{c} \right) \\ &= -\frac{4\pi(f_0 + f_\tau)}{c} \sum_{i=2}^{+\infty} \frac{i-1}{i} C_i \left[ P\left( f_\eta + \frac{2\pi(f_0 + f_\tau)R_1}{c} \right) \right]^i \\ &-2\pi \left( f_\eta + \frac{2\pi(f_0 + f_\tau)R_1}{c} \right) \sum_{i=2}^{+\infty} C_i \left[ P\left( f_\eta + \frac{2\pi(f_0 + f_\tau)R_1}{c} \right) \right]^{i-1} \\ &- \frac{\pi f_\tau^2}{K_r} - \frac{4\pi(f_0 + f_\tau)R_0}{c} \end{aligned} \tag{19}$$

## 3. Imaging Algorithm

Based on the high$-$order 2$-$D spectrum proposed above, a novel ultra$-$high$-$resolution imaging algorithm for space$-$borne SAR is presented in this section. The sever azimuth aliasing caused by the spotlight or sliding spotlight mode is eliminated by the two$-$step processing method [28,29]. Then, a range high$-$order phase filter and a linear RCMC filter are applied to fulfill the range preprocessing. Further, a cubic phase function is introduced to eliminate the space$-$variance of the range chirp rate, and a quadratic RCMC filter (after the range SRC) is employed to correct the new RCM generated by the chirp rate

equalization. Finally, the azimuth compression filter is applied to focus the targets, and the azimuth scaling is adopted to avoid the azimuth time domain aliasing caused by the deramp operation. Figure 1 gives the block diagram of the proposed imaging algorithm, and all these parts mentioned above are illustrated in the following:

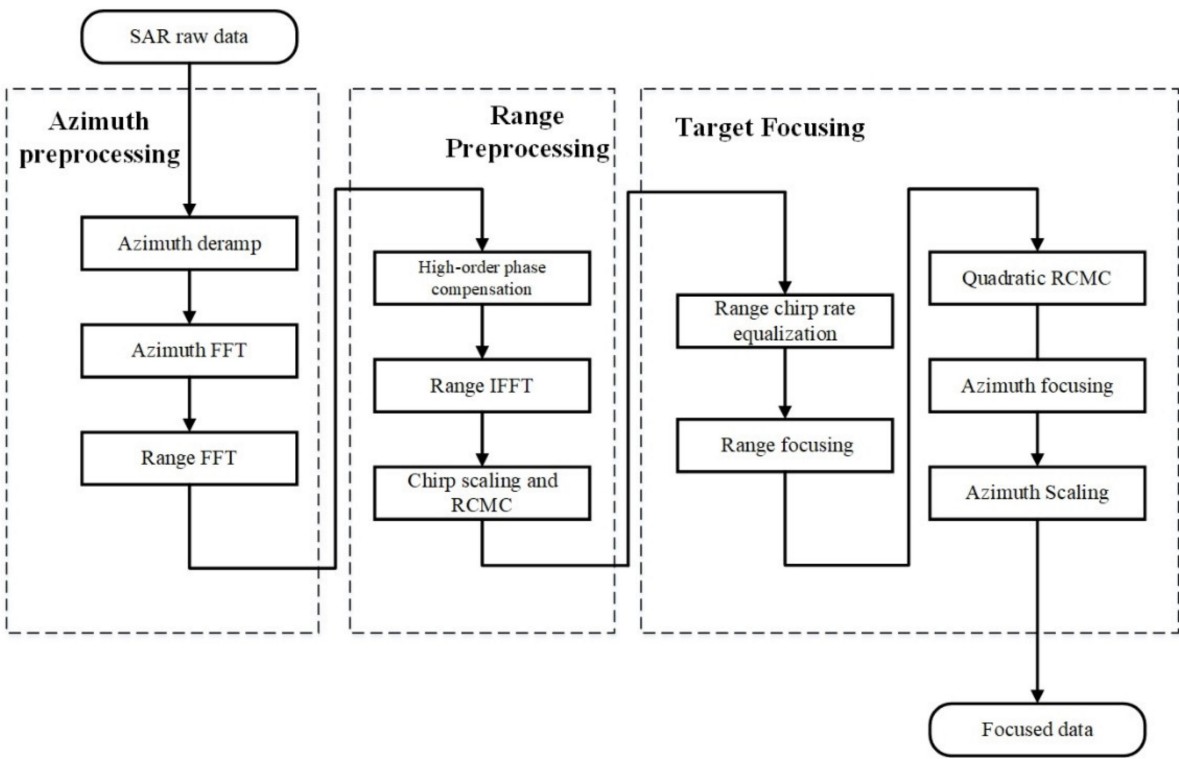

**Figure 1.** Detailed implementation of the proposed algorithm.

### 3.1. Azimuth Preprocessing

The total Doppler bandwidth of an ultra−high−resolution space−borne SAR echo is always far larger than the pulse repetition frequency (PRF), which is limited by the range ambiguity and data volume. Therefore, the signal spectrum suffered from the severe aliasing at the azimuth direction. The time−frequency diagram of the sliding spotlight, which is widely developed in many advanced SAR systems and applied in our simulations, is elaborated in Figure 2. The azimuth bandwidth of the whole scene $B_{total}$ is composed of $B_{steer}$ and $B_{3dB}$, while $B_{steer}$ is rendered by the azimuth antenna steering and $B_{3dB}$ is the antenna beam 3 dB bandwidth. $B_T$ is the single point target's azimuth bandwidth and $T_B$ is the azimuth integration time. As shown in Figure 2, the aliasing is inevitable and must be removed for post−processing.

The two−step processing method is an efficient method and is employed in our algorithm. It utilizes a reference LFM signal with an opposite Doppler chirp rate convoluted with the received signal in the azimuth direction. $B_{steer}$ is removed through this operation and the processed signal can be expressed as Equation (20)

$$s'(\tau, \eta) = s(\tau, \eta) * exp\left\{-j\pi K_{r\_rot}\eta^2\right\} \tag{20}$$

while $K_{r\_rot}$ is the azimuth chirp rate introduced by the antenna steering and $*$ is the convolution operator. The sketches of the spectrum, whether processed by this method or not, are elaborated in Figure 3, which demonstrate the azimuth wrapping is eliminated clearly in Figure 3b.

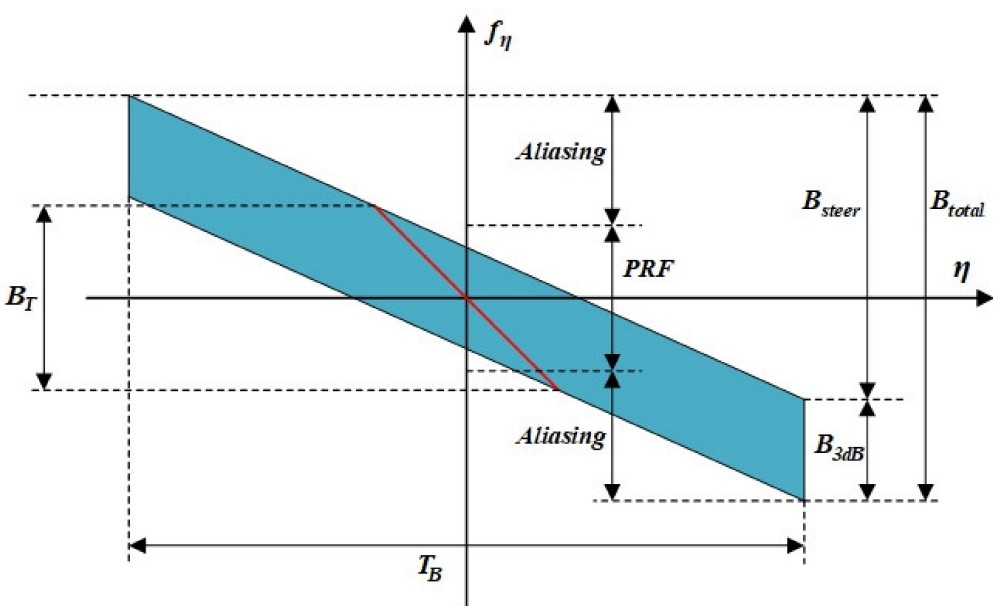

**Figure 2.** Time−frequency diagram of the sliding spotlight SAR.

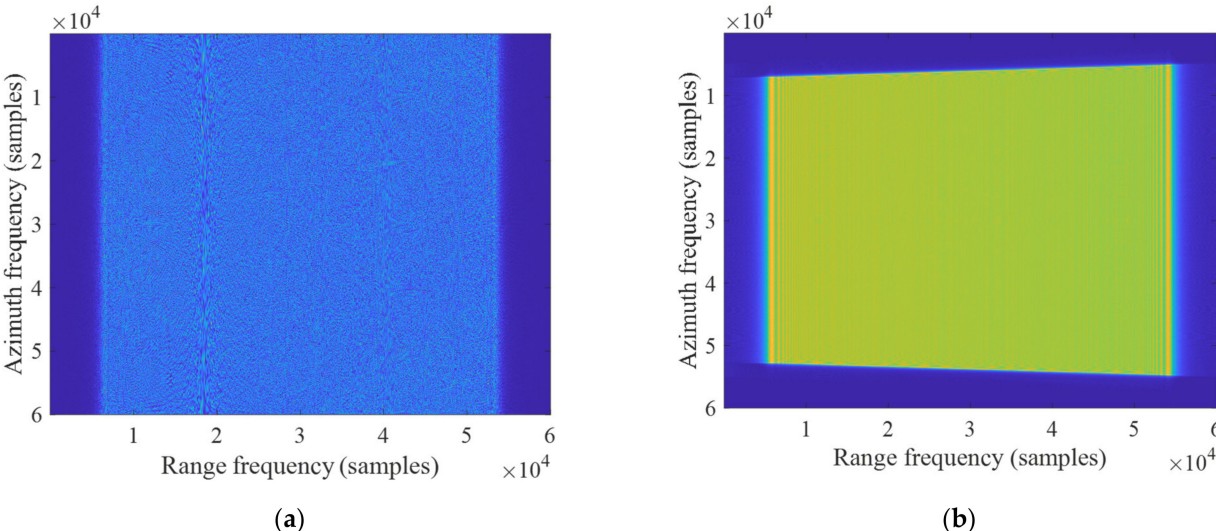

**Figure 3.** The 2−D spectrum of the echo whether the two−step processing is applied. (**a**) The original spectrum. (**b**) the unwrapped spectrum.

In addition, the convolution operation requires many computing resources. To simplify the calculation, it can be divided into three parts [30]: the two complex multiplications and one Fast Fourier Transform (FFT) as Equation (21)

$$s_{deramp}(\tau, \eta_1) = exp\left\{-j\pi\frac{\eta_1}{K_{r\_rot}}\right\} \times IFFT\left[s(\tau, \eta)exp\left(-j\pi K_{r\_ref}\eta^2\right)\right] \quad (21)$$

while $\eta_1 = K_{r\_rot}\eta$, and the *PRF* is increased equivalently as

$$PRF_{new} = \frac{N_a K_{r\_rot}}{PRF} \quad (22)$$

while $N_a$ is the azimuth sampling number. The unwrapped 2−D dimension signal spectrum can be obtained based on the time−frequency properties of the Fourier transform as Equations (23) and (24).

$$S_{deramp}(f_\tau, f_\eta) = \sigma_0 \omega_r(f_\tau) \omega_a(f_\eta - f_{\eta c}) exp\{j\Phi_{deramp}(f_\tau, f_\eta)\} \tag{23}$$

$$\begin{aligned}
\Phi_{deramp}(f_\tau, f_\eta) = &-\frac{4\pi(f_0+f_\tau)}{c} \sum_{i=2}^{+\infty} \frac{i-1}{i} C_i \left[ P\left(f_\eta + \frac{2\pi(f_0+f_\tau)R_1}{c}\right)\right]^i \\
&- 2\pi\left(f_\eta + \frac{2\pi(f_0+f_\tau)R_1}{c}\right) \sum_{i=2}^{+\infty} C_i \left[P\left(f_\eta + \frac{2\pi(f_0+f_\tau)R_1}{c}\right)\right]^{i-1} \\
&- \frac{\pi f_\tau^2}{K_r} - \frac{4\pi(f_0+f_\tau)R_0}{c} - \frac{\pi f_\eta^2}{K_{r\_rot}}
\end{aligned} \tag{24}$$

### 3.2. High−Order Phase Compensation

The phase mathematical expression of the 2−D spectrum is portrayed in Equation (24), which embodies significant range−azimuth coupling. To illustrate validly, the phase of spectrum can be decomposed by the Taylor series with $f_\tau$ and expressed as Equation (25).

$$\Phi_{deramp}(f_\tau, f_\eta) = \sum_{i=0}^{+\infty} D_i(f_\eta) f_\tau^i \tag{25}$$

while $D_0(f_\eta)$ is the term of azimuth Doppler frequency modulation, which is recognized as the LFM term in most instances. $D_1(f_\eta)$ includes the information of the RCM for each target, and $D_2(f_\eta)$ corresponds to the range modulation. Their approximate expression is given in Appendix A, and the remaining higher order terms donate the high−order range–azimuth coupling.

Many frequency algorithms are based on the first−three order of the spectrum and focus less on the others, such as the RDA and CSA. However, the negligence of the high−order term does not work for the high−resolution case. The range distortion caused by it is elaborated in Figure 4. [24] gives an analysis of phase error introduced by each order term of the resolution of 1 m, and concludes that the sixth order coupling phase term still requires considering. Through this, it is evident that the higher resolution required, the higher phase terms need to be considered.

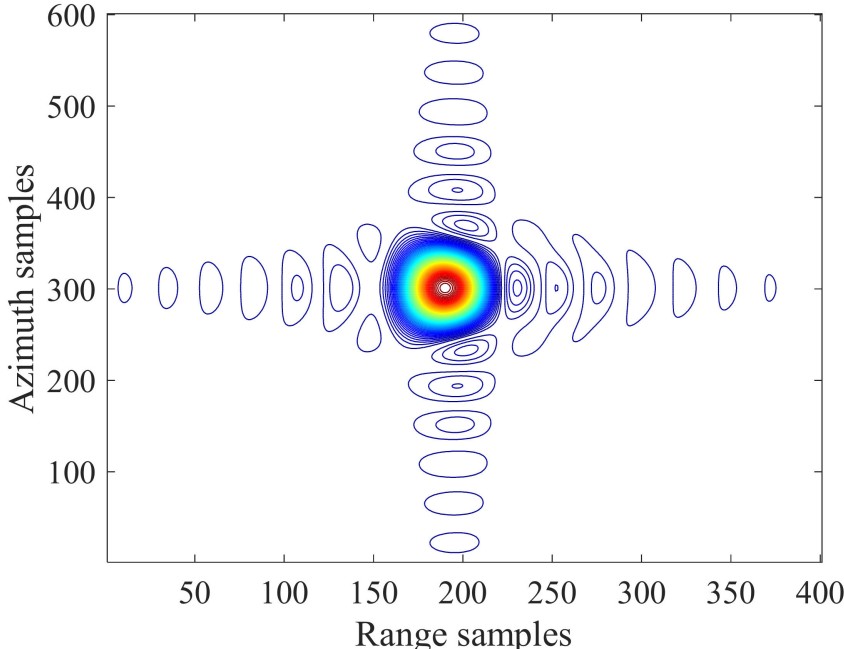

**Figure 4.** The range distortion caused by the high−order phase term.

To obtain the well−focused targets, the high−order phase term is compensated in this paper, which can be calculated by

$$\Phi_{high\_order}\left(f_\tau, f_\eta\right) = \Phi_{deramp}\left(f_\tau, f_\eta\right) - \left(D_0\left(f_\eta\right) + D_1\left(f_\eta\right)f_\tau + D_2\left(f_\eta\right)f_\tau^2\right) \tag{26}$$

and the compensated 2−D spectrum can be achieved by

$$\begin{aligned}
S_c\left(f_\tau, f_\eta\right) &= S_{deramp}\left(f_\tau, f_\eta\right) \times exp\left\{-j\Phi_{high_{order}}\left(f_\tau, f_\eta\right)\right\} \\
&= \sigma_0\omega_r(f_\tau)\omega_a\left(f_\eta - f_{\eta c}\right)exp\left\{j\left(D_0\left(f_\eta\right) + D_1\left(f_\eta\right)f_\tau + D_2\left(f_\eta\right)f_\tau^2\right)\right\}
\end{aligned} \tag{27}$$

Transforming the signal to the range time domain by a range inverse fast Fourier transform (IFFT), at last, the compensated signal in the RD domain is achieved by

$$S_c\left(\tau, f_\eta\right) = \sigma_0\omega_r\left(\tau - E_2\left(f_\eta\right)\right)\omega_a\left(f_\eta - f_{\eta c}\right)exp\left\{jE_0\left(f_\eta\right)\right\}exp\left\{-jE_1\left(f_\eta\right)\left(\tau - E_2\left(f_\eta\right)\right)^2\right\} \tag{28}$$

while

$$\begin{cases}
E_0\left(f_\eta\right) = D_0\left(f_\eta\right) \\
E_1\left(f_\eta\right) = \dfrac{\pi^2}{D_2\left(f_\eta\right)} \\
E_2\left(f_\eta\right) = -\dfrac{D_1\left(f_\eta\right)}{2\pi}
\end{cases} \tag{29}$$

*3.3. Linear RCMC*

As the principle of chirp scaling, the chirp signal can adjust the phase center by multiplying itself with another chirp signal. By these means, the different RCM produced by each target, which is located in different range gates, can be adjusted to the same, and this is named differential RCMC in many papers.

For the expression, according to the Equations (28) and (29), the range chirp rate $br\left(f_\eta\right)$ and range cell migration(RCM) $R_{rd}\left(f_\eta\right)$ in the RD domain can be derived as

$$br\left(f_\eta\right) = \frac{E_1\left(f_\eta\right)}{\pi} \tag{30}$$

$$R_{rd}\left(f_\eta\right) = \frac{cE_2\left(f_\eta\right)}{2} \tag{31}$$

Therefore, the chirp scaling factor is

$$\Phi_{cs}\left(\tau, f_\eta\right) = exp\left\{-j\pi br\left(f_\eta, R_{ref}\right)Cs\left(f_\eta\right)\left(\tau - \tau_{ref}\left(f_\eta\right)\right)^2\right\} \tag{32}$$

where

$$Cs\left(f_\eta\right) = \frac{R\left(f_\eta, R_{ref}\right)}{R_0} - 1 = \frac{cE_2\left(f_\eta, R_{ref}\right)}{2R_{ref}} - 1 \tag{33}$$

$$\tau_{ref}\left(f_\eta\right) = \frac{2R_{rd}\left(f_\eta, R_{ref}\right)}{c} \tag{34}$$

$R_{ref}$ represented the referenced slant range of the referenced range gate. The differential RCMC is achieved by multiplying $\Phi_{cs}\left(\tau, f_\eta\right)$ with $S_c\left(\tau, f_\eta\right)$ in the RD domain.

$$\begin{aligned}
S_{cs}\left(\tau, f_\eta\right) &= S_c\left(\tau, f_\eta\right) \times exp\left\{j\Phi_{cs}\left(\tau, f_\eta\right)\right\} \\
&= \sigma_0\omega_r\left(\tau - E_2\left(f_\eta\right)\right)\omega_a\left(f_\eta - f_{\eta c}\right)exp\left\{jE_0\left(f_\eta\right)\right\} \\
&\times exp\left\{-j\pi br\left(f_\eta\right)\left[1 + Cs\left(f_\eta\right)\right]\left[\tau - \frac{E_2\left(f_\eta\right) + Cs\left(f_\eta\right)\tau_{ref}\left(f_\eta\right)}{1 + Cs\left(f_\eta\right)}\right]^2\right\} \\
&\times exp\left\{j\pi br\left(f_\eta\right)\left[\frac{\left(E_2\left(f_\eta\right) + Cs\left(f_\eta\right)\tau_{ref}\left(f_\eta\right)\right)^2}{1 + Cs\left(f_\eta\right)} - E_2^2\left(f_\eta\right) - Cs\left(f_\eta\right)\tau_{ref}^2\left(f_\eta\right)\right]\right\}
\end{aligned} \tag{35}$$

After the chirp scaling operation is finished, the remaining RCMC can be achieved together by the RCM of the referenced range gate and is always realized by a complex multiplication in the 2−D frequency domain. According to the time−frequency properties of the Fourier transform, the remaining RCMC filter in the 2−D frequency domain can be expressed as Equation(36)

$$H_{rcmc}\left(f_\tau, f_\eta\right) = exp\left\{ j\frac{4\pi f_\tau R_{ref} Cs\left(f_\eta\right)}{c} \right\} \tag{36}$$

and the signal in RD domain after finished the linear RCMC is

$$
\begin{aligned}
S_{rcmc}\left(\tau, f_\eta\right) &= IFFT\left[FFT\left[S_{cs}\left(\tau, f_\eta\right), \tau\right] \times H_{rcmc}\left(f_\tau, f_\eta\right), f_\tau\right] \\
&= \sigma_0 \omega_r\left(\tau - \frac{2R}{c}\right)\omega_a\left(f_\eta - f_{\eta_c}\right) \\
&\times exp\left\{-j\pi br\left(f_\eta\right)\left[1 + Cs\left(f_\eta\right)\right]\left[\tau - \frac{2R}{c}\right]^2\right\} \\
&\times exp\left\{jE_0\left(f_\eta\right)\right\}exp\left\{j\pi br\left(f_\eta\right)\left[\frac{\left(E_2\left(f_\eta\right)+Cs\left(f_\eta\right)\tau_{ref}\left(f_\eta\right)\right)^2}{1+Cs\left(f_\eta\right)} - E_2^2\left(f_\eta\right) - Cs\left(f_\eta\right)\tau_{ref}^2\left(f_\eta\right)\right]\right\}
\end{aligned}
\tag{37}
$$

while the *R* is the referenced slant range for each range gate.

### 3.4. Chirp Rate Equalization and Quadratic RCMC

The chirp rate is space−variant in the RD domain. This variance is always regarded as a system error for the traditional imaging algorithm and neglected because of the low resolution. However, for the ultra−high−resolution case, this is sufficient to lead the targets at the edge to defocus seriously. Based on the first exponential term in Equation (37), the range chirp rate in the RD domain can be expressed as $K_r\left(f_\eta, R\right)$

$$K_r\left(f_\eta, R\right) = br\left(f_\eta, R\right)\left[1 + Cs\left(f_\eta\right)\right] \tag{38}$$

while it is assumed to vary linearly with *R* in this paper like Equation (39)

$$K_r\left(f_\eta, R\right) = K_r\left(f_\eta, R_{ref}\right) + A\left(f_\eta\right)\left(R - R_{ref}\right) \tag{39}$$

and $A\left(f_\eta\right)$ is the rate of the $K_r\left(f_\eta, R\right)$. So, this quadratic exponential term can be re−expressed as

$$
\begin{aligned}
&exp\left\{-j\pi K_r\left(f_\eta, R\right)\left[\tau - \frac{2R}{c}\right]^2\right\} \\
&= exp\left\{-j\pi\left[K_r\left(f_\eta, R_{ref}\right) + A\left(f_\eta\right)\left(R - R_{ref}\right)\right]\left[\tau - \frac{2R}{c}\right]^2\right\}
\end{aligned}
\tag{40}
$$

Based on [26], it is feasible to apply a cubic phase perturbation factor, given in Equation (41), to equalize the chirp rate, like the chirp scaling principle

$$H_{ec}\left(f_\eta\right) = exp\left\{-j\pi\frac{A\left(f_\eta\right)}{3}\left[\tau - \frac{2R_{ref}}{c}\right]^3\right\} \tag{41}$$

the quadratic term after multiplied with Equation (41) becomes Equation (42).

$$
\begin{aligned}
&exp\left\{ -j\pi K_r\left(f_\eta, R\right)\left[\tau - \tfrac{2R}{c}\right]^2 \right\} \times H_{ec}\left(f_\eta\right)\\
&= exp\left\{ -j\pi \tfrac{A\left(f_\eta\right)}{3}\left(\tau - \tfrac{2R}{c}\right)^3 \right\}\\
&\times exp\left\{ -j\pi K_r\left(f_\eta, R_{ref}\right)\left[\tau - \tfrac{2R}{c}\right]^2 \right\}\\
&\times exp\left\{ -j\pi A\left(f_\eta\right)\left(\tfrac{2R}{c} - \tfrac{2R_{ref}}{c}\right)^2\left(\tau - \tfrac{2R}{c}\right) \right\}\\
&\times exp\left\{ -j\pi \tfrac{A\left(f_\eta\right)}{3}\left(\tfrac{2R}{c} - \tfrac{2R_{ref}}{c}\right)^3 \right\}
\end{aligned}
\tag{42}
$$

The chirp rate for each target located at each range gate is equalized to the reference chirp rate in Equation (42), and it can be well−compressed with a match filter Equation (43) in the RD domain accurately. The first exponential term is the cubic phase term, which can be compensated in the RD domain by the POSP, and the last exponential term is constant.

$$
H_{rc}\left(f_\tau, f_\eta\right) = exp\left\{ -j\pi \frac{f_\tau^2}{K_r\left(f_\eta, R_{ref}\right)} \right\}
\tag{43}
$$

The most difficult is the remaining linear term. It will cause the phase center shift in the range time domain and bandwidth shift in the frequency domain. Moreover, a false target will be produced if the bandwidth shifts too much. To explain this phenomenon straightforwardly, Figure 5 shows the false target caused by the spectrum wrapping. The bandwidth of the original spectrum is included in $[-F_s/2 : F_s/2]$, so its compressed result is well−focused and located at the phase center of the chirp signal. However, if the bandwidth shifts out of the sampling frequency, as illustrated in Figure 5a, the energy leakage will occur, and the false target will emerge, as shown in Figure 5b. Therefore, the sampling frequency must be large enough to ensure the bandwidth does not alias after the shift.

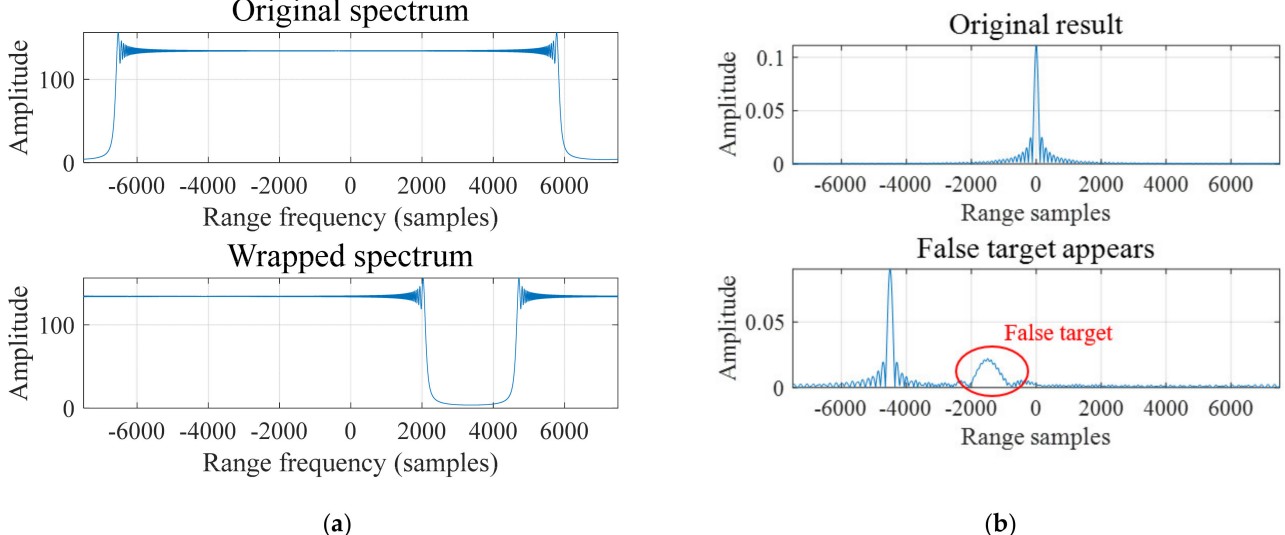

**Figure 5.** (**a**) The original spectrum and the wrapped spectrum due to the linear term. (**b**) The original focused result and the false target formed by wrapping.

Nevertheless, phase center shifting is inevitable in the range time domain. Figure 6a gives a plot of a real part from a chirp signal, adjusted to the chirp rate by the aforementioned cubic phase perturbation factor. The phase center of the original signal is located at the center of the plot. However, it shifted to the left after applying the chirp rate equal-

ization, displayed in Figure 6a below. Their compressed results, focused on the reference chirp rate, are displayed in Figure 6b. The defocusing emerges on the original signal due to the mismatch of the match filter, and the equalized signal reflects the well−focusing. The peaks of each pulse appear in the phase center so that there is a deviation of location between them. When this bias varies along with the azimuth frequency, it manifests as the RCM introduced artificially. Furthermore, due to the shifts corresponding to the square of the range difference, which embodies the third exponential of Equation (40), the newly generated RCM is represented as quadratic and toward the same direction as illustrated in Figure 7.

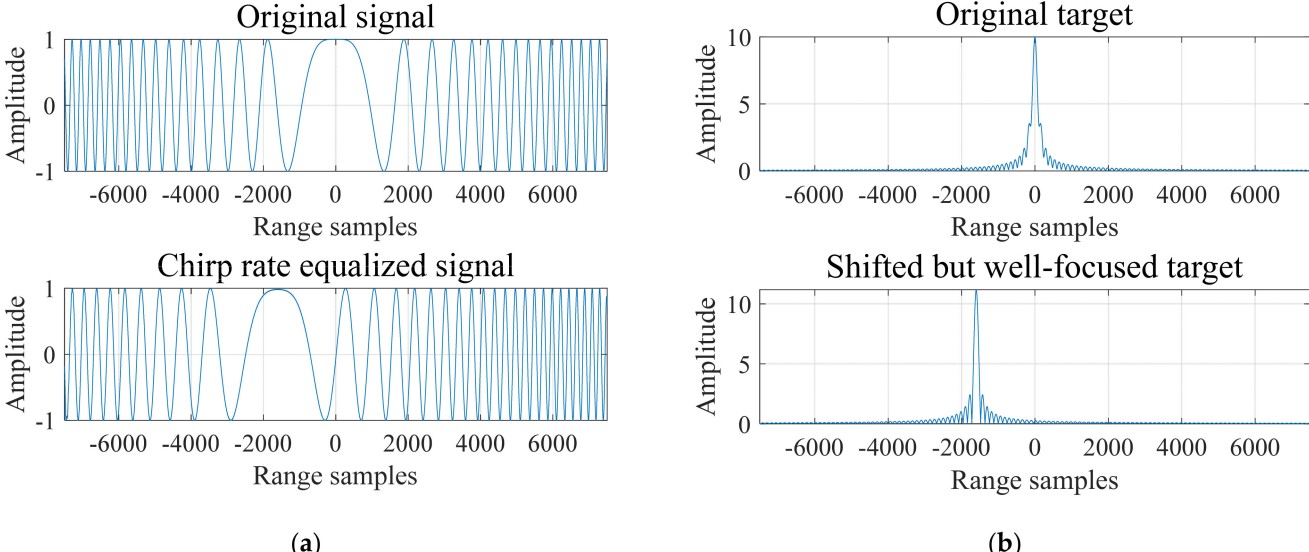

**Figure 6.** (**a**) The real part of the chirp signal before and after the chirp rate equalization. (**b**) the range compression result focused by the reference chirp rate before and after the chirp rate equalization.

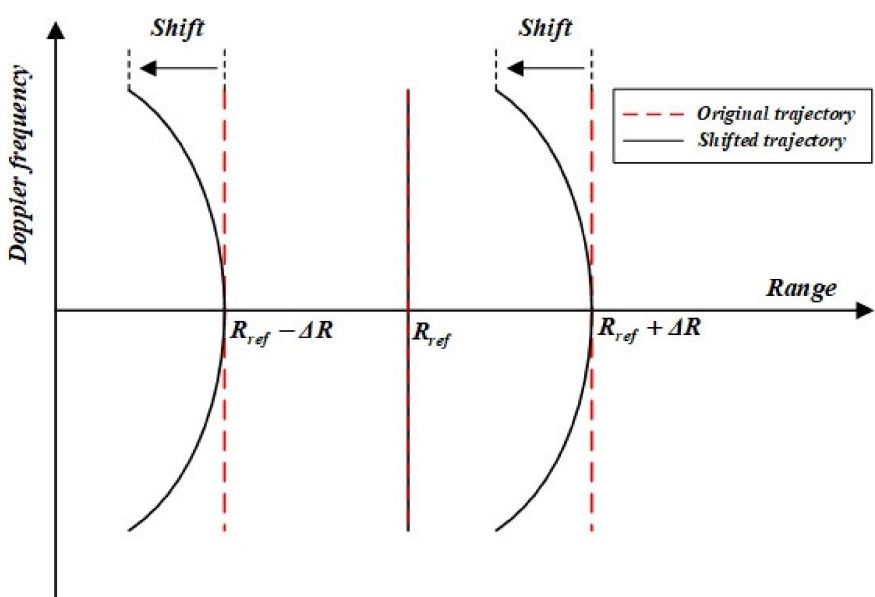

**Figure 7.** The diagram of quadratic RCM.

　　The principle of chirp scaling is no longer applicable for the correction because the quadratic RCM is introduced by itself. To accomplish the RCMC, interpolation is employed in this paper. The shift of each range bin can be calculated in Equation (44) precisely without any high−order term. Therefore, the interpolation kernel is longer, and the RCMC by this

method is more accurate. At last, the expression of the signal after RCMC and residual phase compensation is given in Equation (45).

$$RCM_{quadratic}\left(f_\eta\right) = \frac{A\left(f_\eta\right)\left(\frac{2R}{c} - \frac{2R_{ref}}{c}\right)^2}{2K_r\left(f_\eta, R_{ref}\right)} \tag{44}$$

$$S_{rc}\left(\tau, f_\eta\right) = \sigma_0 A_r\left(\tau - \frac{2R_{ref}}{c}\right)\omega_a\left(f_\eta - f_{\eta c}\right)$$
$$\times exp\left\{jE_0\left(f_\eta\right)\right\}exp\left\{j\pi br\left(f_\eta\right)\left[\frac{\left(E_2\left(f_\eta\right)+Cs\left(f_\eta\right)\tau_{ref}\left(f_\eta\right)\right)^2}{1+Cs\left(f_\eta\right)} - E_2^2\left(f_\eta\right) - Cs\left(f_\eta\right)\tau_{ref}^2\left(f_\eta\right)\right]\right\} \tag{45}$$

while the $A_r(\cdot)$ is the range envelope after range compression and usually as a sinc function.

*3.5. Azimuth Compression and Resampling*

According to the Equation (45), the remaining phase includes two parts. The penultimate term is the azimuth modulation phase mentioned above. The last exponential term is introduced by the chirp scaling operation, both of which need to be compensated accurately. Therefore, the azimuth compression filter can be obtained through the conjugate of these parts.

$$H_{ac}\left(\tau, f_\eta\right) = exp\left\{-j\left[E_1\left(f_\eta\right)\left(\frac{\left(E_2\left(f_\eta\right)+Cs\left(f_\eta\right)\tau_{ref}\left(f_\eta\right)\right)^2}{1+Cs\left(f_\eta\right)} - E_2^2\left(f_\eta\right) - Cs\left(f_\eta\right)\tau_{ref}^2\left(f_\eta\right)\right) + E_0\left(f_\eta\right)\right]\right\} \tag{46}$$

To eliminate the azimuth time domain aliasing caused by the deramp, the azimuth scaling option is necessary to achieve the azimuth resampling. The scaling function is given in Equation (47) as

$$H_{scaling}\left(f_\eta\right) = exp\left\{-j\pi\frac{H_f}{K_{r\_rot}}f_\eta^2\right\} \tag{47}$$

while the $H_f$ is the hybrid factor. After performing the azimuth IFFT, the residual phase is given in Equation (48), and the focused target can be expressed as Equation (49).

$$H_{residual}\left(\eta\right) = exp\left\{-j\pi\frac{K_{r\_rot}}{H_f}\eta^2\right\} \tag{48}$$

$$S_{focused}\left(\tau, \eta\right) = IFFT\left[IFFT\left[S_{rc}\left(\tau, f_\eta\right) \times H_{ac}\left(\tau, f_\eta\right) \times H_{scaling}\left(f_\eta\right), f_\eta\right] \times H_{residual}\left(\eta'\right), \eta'\right]$$
$$= \sigma_0 A_r\left(\tau - \frac{2R_{ref}}{c}\right)A_a\left(\eta - \eta_c\right) \tag{49}$$

while the $A_a(\cdot)$ is the azimuth envelope function.

**4. Discussion and Simulation Results**

The discussion and experiment results of simulation are presented in this section. To verify the proposed algorithm, the main simulation parameters are given in Table 1.

**Table 1.** Simulation Parameters.

| Description | Value | Units |
|---|---|---|
| Orbit Parameters | | |
| Semi−major | 514 | km |
| Eccentricity | 0.0011 | − |
| Inclination | 98 | deg |
| Longitude of ascend note | 0 | deg |
| Argument of perigee | 90 | deg |
| Radar Parameters | | |
| Carrier frequency | 9.6 | GHz |
| Bandwidth | 1.25 | GHz |
| Sampling frequency | 1.5 | GHz |
| Look angle | 30 | deg |
| Antenna width | 0.7 | m |
| Antenna Length | 6 | m |
| Target Parameters of Scene Center | | |
| Azimuth resolution | 0.25 | m |
| Hybrid factor | 0.075 | − |
| $R_0$ of scene center | 593.42 | km |
| $f_d$ of scene center | 4.969 | Hz |
| $f_r$ of scene center | 5909.132 | Hz/s |
| $f_{r3}$ of scene center | 0.019894 | Hz/s$^2$ |
| $f_{r4}$ of scene center | 2.765661 | Hz/s$^3$ |

*4.1. The Discussion of The Polynomial Range Model and High−Order 2−D Spectrum Analyses*

The infinite range model is unachievable in engineering. Therefore, the truncation of it is necessary. There are two infinite series while the range model deduction. First is the range vector illustrated in Equation (1). Each order relative motion parameter vector is required in this expression. However, the higher−order term is non−essential because the illumination time is limited. Figure 8 shows the phase error calculated by the modulus of the low−order truncated range vector and the real range history, and the red line annotates the safe line under the azimuth phase error criterion of 0.25 π. As portrayed in Figure 8, the first four−order relative motion parameters can fit the real range history in 20 s. This is enough to meet the requirement of the ultra−high−resolution imaging for now, and for some time to come.

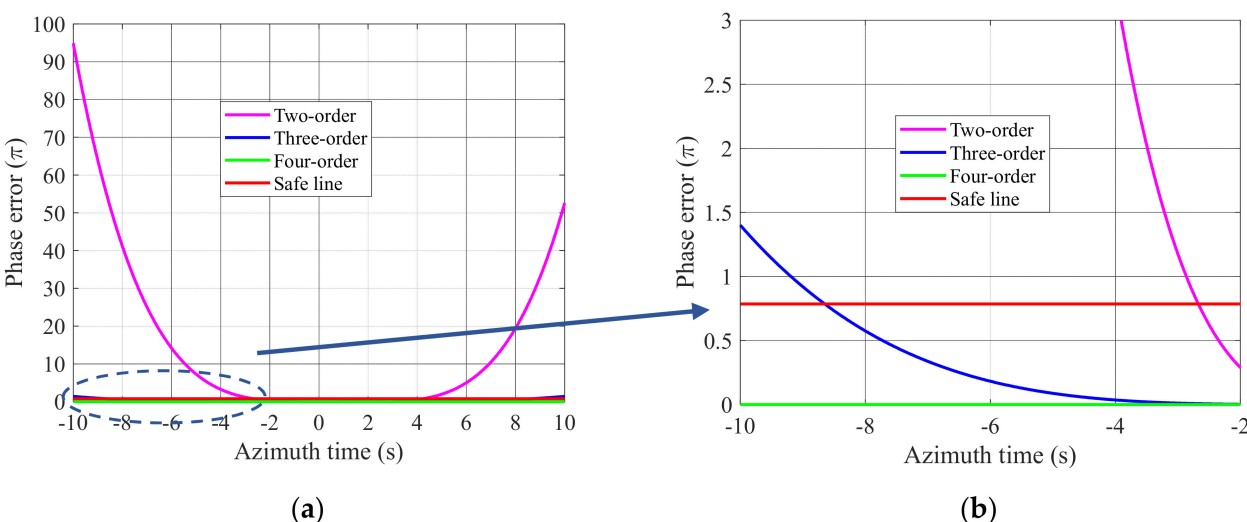

**Figure 8.** (**a**) The phase error introduced by the truncated range vector. (**b**) the detail view of the circled place.

The second infinite series is the scalar expression of the range model acquired by the Taylor expansion. Similarly, it is not necessary and impossible to apply all coefficients to obtain the no error range history. The first several order approximations will offer acceptable performance and reduce the calculation in engineering. Figure 9 portrays the approximation results based on the first−four order relative motion parameters in the same manner as above. Moreover, the only even−order truncated results are elaborated in the plot because the odd−order coefficients are calculated close to zero through the simulation parameters presented in Table 1. The six−order approximated range model is enough for the 20 s integration time, while the requirement of the ultra−high−resolution imaging is already achieved. However, the curve has an apparent upward trend after −6 s, so to satisfy the development of the SAR system in the future, the eight−order range model will be a better choice, and the remaining experiments are based upon it.

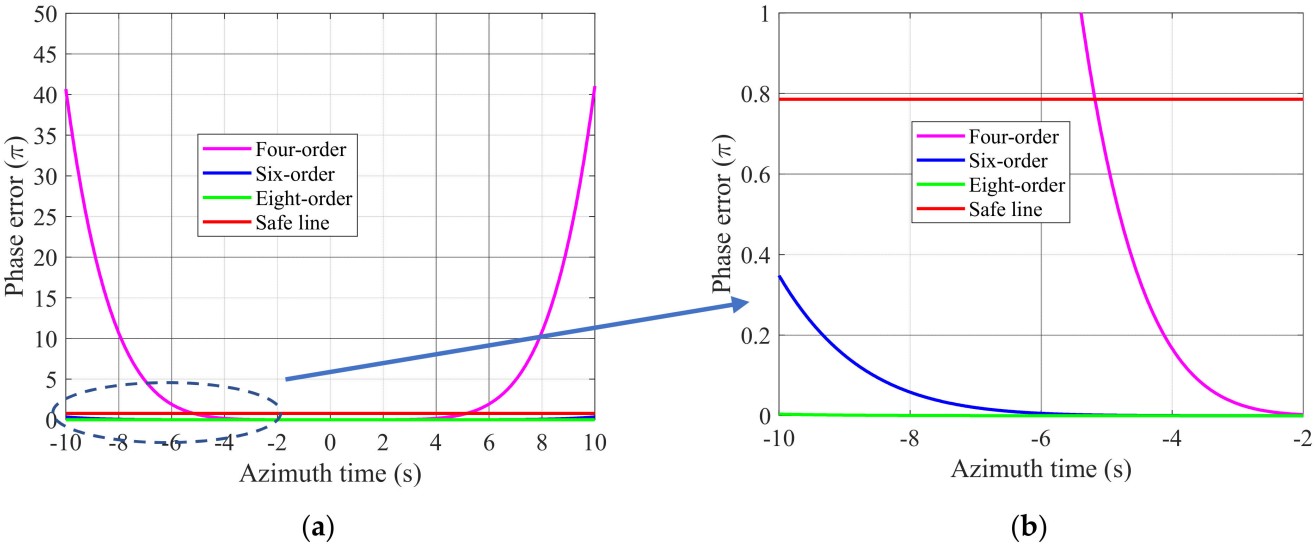

**Figure 9.** (**a**) The phase error introduced by the truncated range model. (**b**) the detail view of the circled place.

Furthermore, we aimed to verify the accuracy of the high−order 2−D spectrum generated by the Lagrange inversion. The peak sidelobe ratio (PSLR) and the normalized impulse response width (IRW) measured from the point targets, compressed by the calculated spectrum directly based on the different order truncated Lagrange inversion, are presented in Figure 10. Obviously, the three−order Lagrange inversion is far from reaching the requirement of ultra−high−resolution imaging. Serious defocusing has emerged since the 0.8 m of the azimuth resolution. In addition, the four− and five−order inversions exhibit the near−precision of the derived spectrum because the fifth−order coefficient is too small to affect the precision, which is almost the same as the sixth−order parameter in this experiment. Moreover, they are all invalid since the azimuth resolution is higher than 0.17 m, where the spectrum obtained by the six−order inversion still performs well. Similarly, the accuracy of the 2−D spectrum can be improved by increasing the order of the Lagrange inversion to approach the actual accuracy like the Taylor series, and the six−order inversion is employed in the next simulation experiment to calculate the 2−D spectrum.

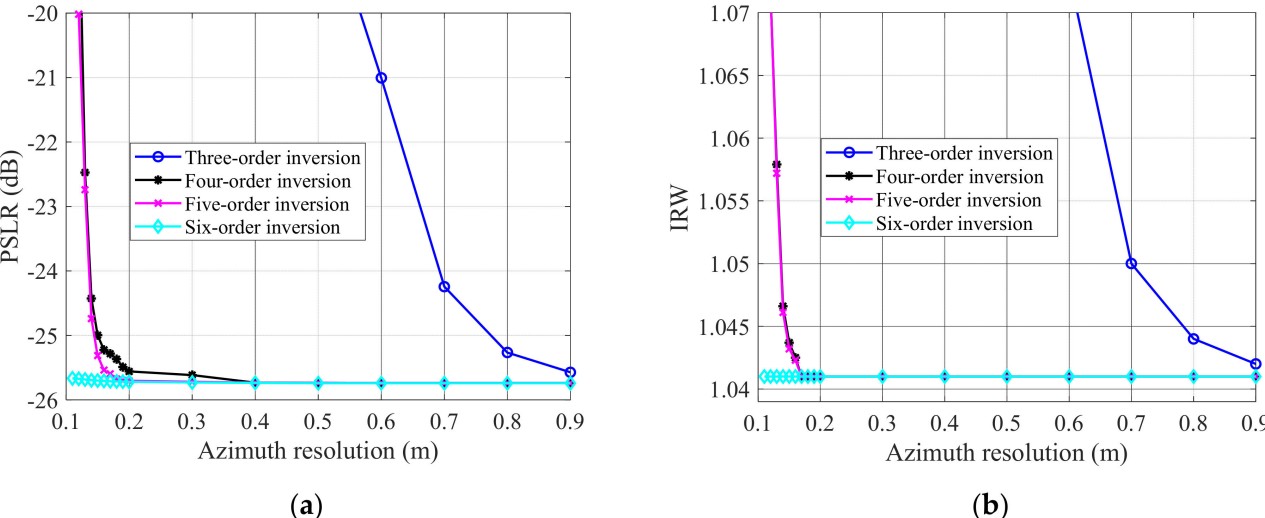

**Figure 10.** (**a**). PSLR of the compressed point targets. (**b**). IRW of the compressed point targets.

*4.2. Image Quality Evaluation and Analysis*

The simulated scene is shown in Figure 11, the nine point targets arranged evenly along with the range, and azimuth in 6 km and 2 km as illuminated in the sliding spotlight mode. The main simulation parameters are given in Table 1. To fit the engineering applications, all simulation results are weighted by the 1/3 simplified Taylor window both in range and azimuth, which can reduce the sidelobe level and avoid flooding the weak targets [31].

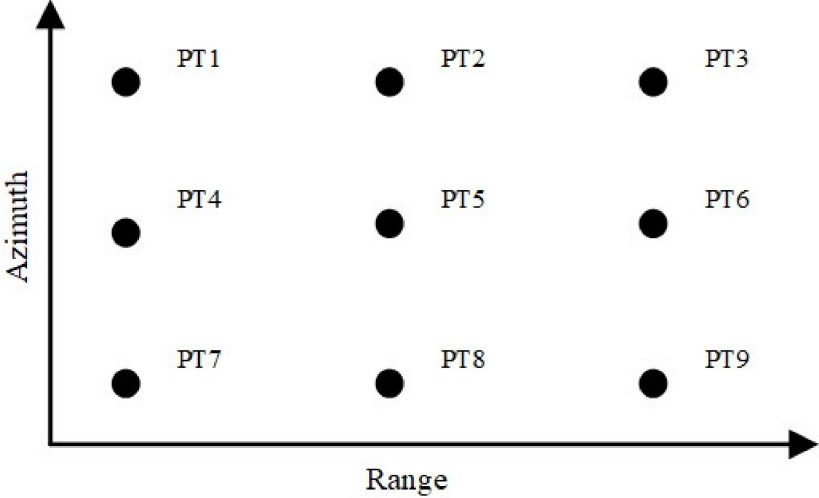

**Figure 11.** The ground scene of simulation.

Figure 12 illustrates the necessity of the range chirp rate equalization. The diagrams of the PT1, whether it eliminated the chirp rate space variance or not after the range compression in the RD domain, is given in Figures 12a and 12b respectively. As seen in Figure 12a, the mismatch of the match filter is reflected obviously, especially at the edge of the Doppler bandwidth. This leads the results to range defocusing finally. In contrast, Figure 12b shows the influence of the chirp rate equalization achieved by the cubic phase function. The mismatch is eliminated effectively through this method. Conversely, a tiny shift in the range direction corresponding to the Doppler frequency is caused and brings the azimuth distortion for each target in the end. To fulfill the quadratic RCMC, the interpolation is employed, and its result is presented in Figure 12c, in which it can be observed that every shift at each Doppler frequency is removed clearly, and the whole energy of a target is well−compressed.

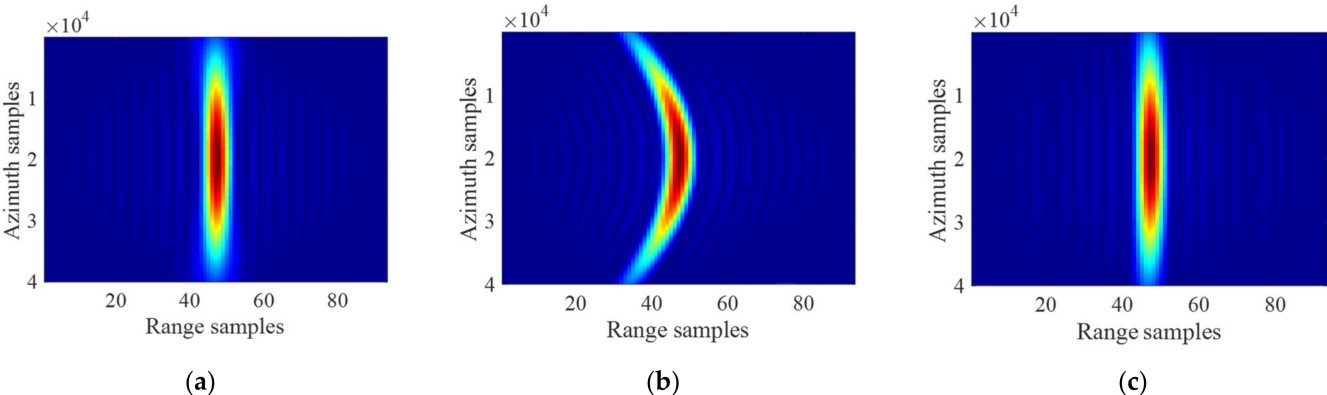

**Figure 12.** The sketch map of the range compression results. (**a**) Compressed signal with a mismatch chirp rate. (**b**) Only the chirp rate equalization is employed and the quadratic RCM appeared. (**c**) Corrected quadratic RCM by means of interpolation.

The interpolated slices of PT1, PT5, and PT9 focused by the proposed algorithm are displayed in Figure 13. In contrast, Figure 14 gives the same targets focused by the conventional CSA, and Figure 15 gives the results from the BPA. The range and azimuth profiles of the PT5 are presented in Figure 15.

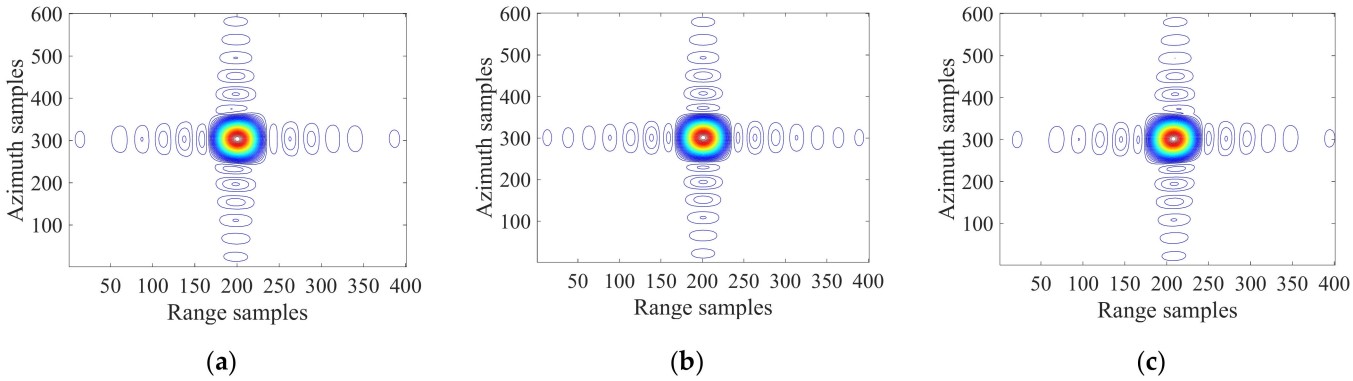

**Figure 13.** Interpolated results of (**a**) PT3, (**b**) PT5, (**c**) PT7 focused by the proposed algorithm.

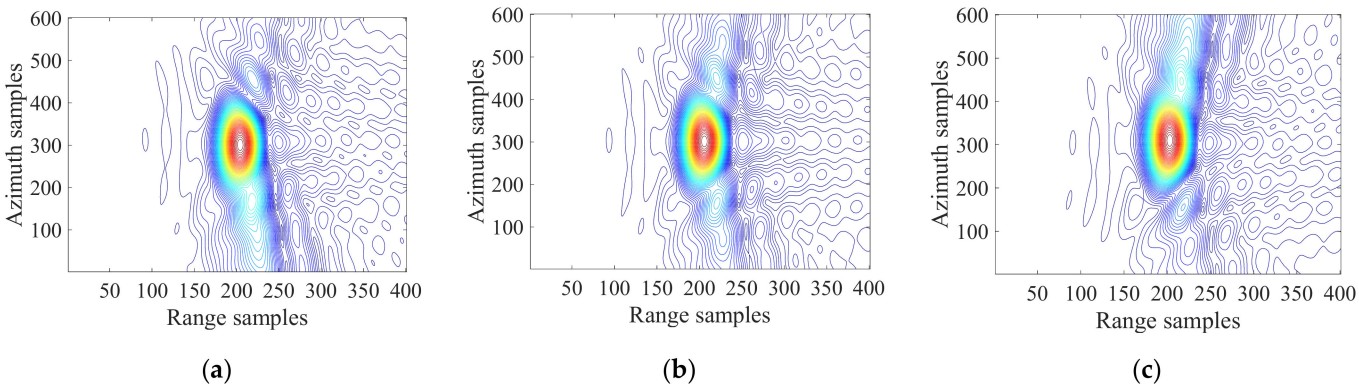

**Figure 14.** Interpolated results of (**a**) PT3, (**b**) PT5, (**c**) PT7 focused by the conventional CSA.

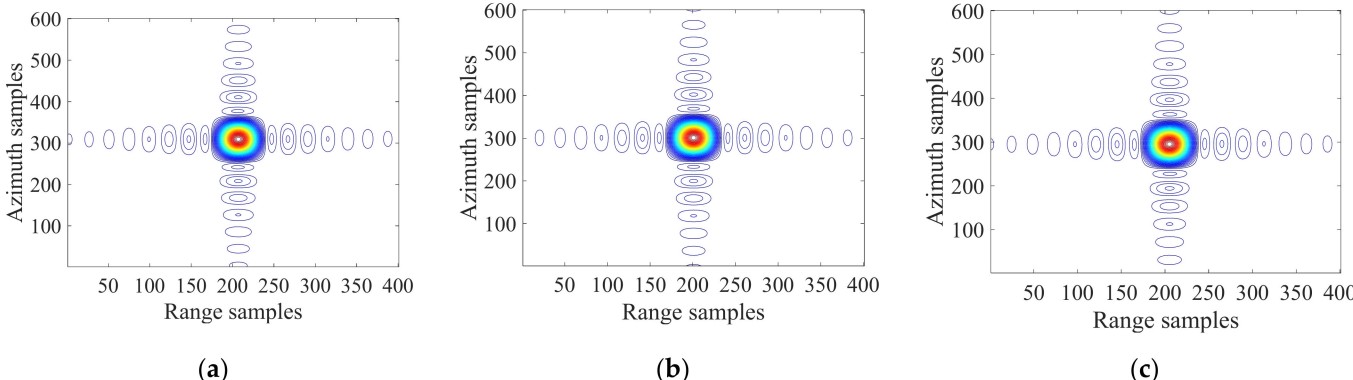

**Figure 15.** Interpolated results of (**a**) PT3, (**b**) PT5, (**c**) PT7 focused by the back−projection algorithm.

As seen from the results processed by CSA, due to the tremendous phase error introduced by the ESRM in the ultra−high−resolution case, each point target suffers from severe degradation, especially at the edge of the azimuth extent. The insufficient precision of the 2−D spectrum computed in CSA causes the rise of PSRL and the widening of IRW. In Figure 16, the profile cannot be recognized as the sinc function.

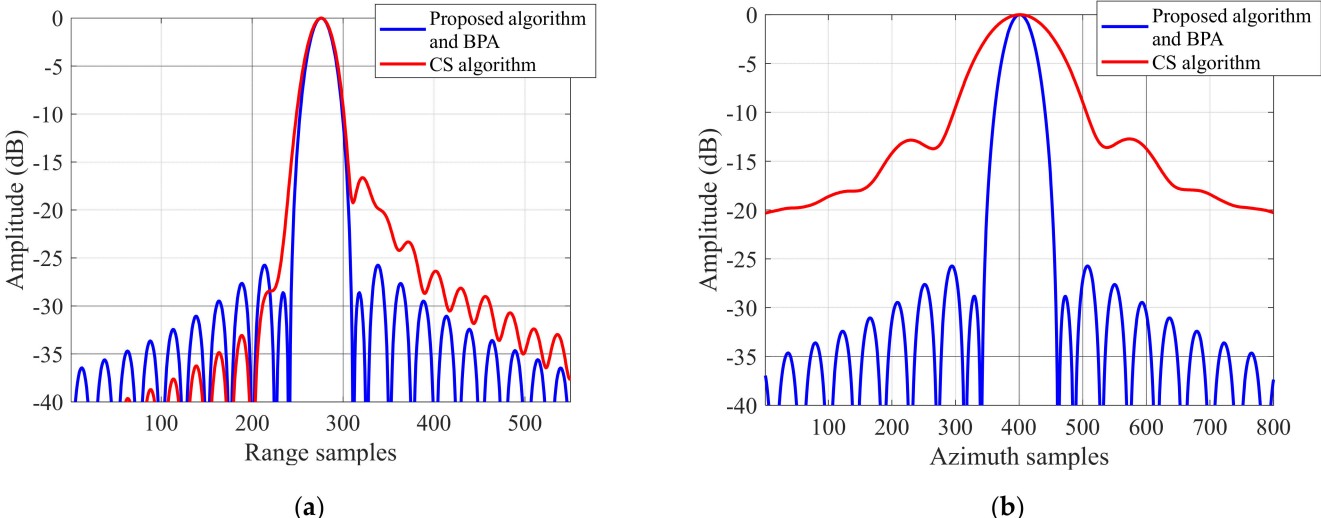

**Figure 16.** The (**a**) range and (**b**) azimuth profile of PT5 focused by the proposed algorithm, BPA, and the conventional CSA.

Conversely, the results from the proposed algorithm, as presented in Figure 13, indicate that with the more accurate range model and the 2−D spectrum based on it, the focusing performance is dramatically improved. Every point target on the scene is well−focused. The results from the BPA are displayed in Figure 15 as a reference. The well−focused targets validate the high imaging accuracy of the time domain algorithm. Compared with Figure 13, two imaging algorithms have similar performances in image quality, but the proposed algorithm only requires eight times FFT, seven times complex multiplication, and one interpolation, which is far more efficient than the BPA.

To better quantify the image quality, the point target analysis results are listed in Table 2. The theoretical resolution, weighted by the 1/3 simplified Taylor window, is calculated in

$$\begin{cases} \rho_r = 1.09 \frac{c}{2B_r} \\ \rho_a = 1.09 \frac{L_a H_f}{2} \end{cases} \tag{50}$$

where $L_a$ is the antenna length and $H_f$ is the hybrid factor.

**Table 2.** Image quality analysis of point targets in small scene obtained by the proposed algorithm.

| Target No. | Range | | | Azimuth | | |
|---|---|---|---|---|---|---|
| | $\rho_r$(m) | PSLR(dB) | ISLR(dB) | $\rho_a$(m) | PSLR(dB) | ISLR(dB) |
| 1 | 0.1357 | −25.856 | −20.807 | 0.2433 | −25.901 | −21.327 |
| 2 | 0.1348 | −25.753 | −20.529 | 0.2427 | −25.729 | −21.123 |
| 3 | 0.1357 | −25.968 | −20.845 | 0.2437 | −25.789 | −21.035 |
| 4 | 0.1352 | −25.725 | −20.655 | 0.2428 | −25.721 | −21.225 |
| 5 | 0.1349 | −25.755 | −20.571 | 0.2427 | −25.737 | −21.195 |
| 6 | 0.1352 | −25.793 | −20.595 | 0.2428 | −25.725 | −21.228 |
| 7 | 0.1356 | −25.894 | −20.736 | 0.2435 | −25.897 | −21.050 |
| 8 | 0.1348 | −25.764 | −20.518 | 0.2427 | −25.729 | −21.138 |
| 9 | 0.1356 | −25.930 | −20.815 | 0.3435 | −25.794 | −21.322 |

It can be observed from Table 2 that the deterioration of the IRW is less than 1%, both in range and azimuth direction. Meanwhile, the PSRL degradation in the table is less than 2%. These promising results prove the proposed algorithm has remarkable performance and can meet the requirement of the ultra−high−resolution imaging for the space−borne SAR for the present and future.

*4.3. Geolocation Analysis*

The information of geolocation, which is widely used in the post−processing of SAR images, is another criterion for evaluating the performance of the imaging algorithm. Table 3 gives the deviation of the positioning results for every focused target in the image scene, while the targets are located in the WGS84 coordinate system. These results indicate that the deviation at each coordinate axis is less than $10^{-2}$ m, proving the proposed algorithm's high position accuracy.

**Table 3.** The deviation of the focused targets in WGS84.

| Target No. | X (m) | Y(m) | Z(m) |
|---|---|---|---|
| 1 | 0.000013 | −0.000237 | 0.000134 |
| 2 | 0.000013 | 0.000300 | −0.001480 |
| 3 | −0.000013 | 0.000232 | −0.000029 |
| 4 | 0.000013 | −0.000248 | −0.000050 |
| 5 | 0.000000 | 0.000000 | 0.000000 |
| 6 | −0.000013 | 0.000211 | 0.000043 |
| 7 | 0.000013 | −0.000249 | 0.000029 |
| 8 | 0.000013 | −0.000456 | 0.002241 |
| 9 | −0.000013 | 0.000244 | 0.000139 |

*4.4. The Discussion of the LEO SAR Motion Error and Autofocus Algorithms*

As can be observed from the infinite polynomial range model, the exact relative motion parameters are required for every order. Nevertheless, the parameters error, especially for the higher order, is inevitable in engineering and degrades the image quality. Motion compensation is a common method of handling this problem. It relies on the inertial navigation system installed on the SAR load to record the movement of the radar and compensate for the residual phase through the stored data. This method requires very high accuracy of the inertial navigation system, which increases the system's final cost.

Autofocus is another way to solve the problem. This algorithm is driven by the echo or image data without any auxiliary equipment. The phase gradient autofocus (PGA) [32] algorithm is the most widely used autofocus method, and the maximum likelihood estimation is employed in the kernel to calculate the phase gradient based on the strongest scattering point in the defocused SAR image, and the estimated phase error gradient compensates the phase error. It is worth noting that PGA can eliminate any order of phase error so that it has a good performance in the high−resolution space−borne SAR. With the

development of deep learning, many autofocus algorithms based on the neural network are proposed now [33,34]. They treat the entire imaging process as a linear transformation of the SAR signal and use the neural network to accomplish the linear mapping. Under certain conditions, they have a similar or even better performance than the classic auto-focus algorithm, but they still have some limitations, such as only being applied on the polynomial form phase error [34] and the vast number of samples required while the neural network is training.

## 5. Conclusions

In this paper, a novel 2−D spectrum required imaging algorithm for the ultra−high−resolution space−borne SAR is proposed. To achieve the accuracy of ultra−high−resolution, a high−precision high−order 2−D spectrum is illustrated in detail at the beginning. The infinite polynomial range model is introduced to the 2−D spectrum analysis, and the expression of the new spectrum is derived from it through the Taylor expansion and Lagrange inversion. In addition, the novel imaging algorithm kernel is presented. The azimuth spectrum aliasing caused by the antenna rotation of the sliding spotlight mode is eliminated by the two−step processing. Moreover, to keep the symmetry on the range direction of the point target's impulse response, the high−order phase term of the echo's spectrum is compensated through the derived new spectrum expression. The linear RCM is corrected through the principle of chirp scaling. The space variance of the chirp rate in the range direction is non−negligible in the ultra−high−resolution case. To solve this problem, a cubic phase function is introduced to equalize the chirp rate and guarantee the quality of range focusing. Meanwhile, the quadratic RCM introduced by the cubic phase function is corrected by interpolating after the range compression. In the end, the azimuth compression is implemented, and the well−focused targets of the large scene can be obtained.

The accuracy of the new spectrum is discussed with the order of the Taylor and Lagrange series in the experiment in this paper, and it verifies that the new spectrum can achieve the requirement of ultra−high−resolution. Then, the image quality of the algorithm is evaluated by the simulation. As a comparison, the imaging results from the traditional CSA are also presented. From the detailed view, it is obvious that every well−focused target derived from the proposed algorithm is defocused and distorted by the CSA. The table of the PSLR and ISLR, calculated from the well−focused targets, further proved the performance of the proposed algorithm. Additionally, the geolocation analysis is also given in this paper. The high positioning accuracy shown in the table ensures the post−processing of a SAR image. All these results verify the effectiveness of the proposed imaging algorithm.

**Author Contributions:** Conceptualization, L.C.; methodology, T.H. and L.C.; software, T.H.; validation, T.H.; formal analysis, T.H.; investigation, T.H.; resources, T.H.; data curation, T.H.; writing original draft preparation, T.H. and Y.G.; writing—review and editing, T.H., L.C., and P.W.; visualization, Y.G. and L.Z.; supervision, L.C. and P.W.; project administration, L.C.; funding acquisition, L.C. All authors have read and agreed to the published version of the manuscript.

**Funding:** This research was funded by National Natural Science Foundation of China (NNSFC) under Grant No. 61861136008 and Shanghai Aerospace Science and Technology Innovation Fund SAST2020−038.

**Data Availability Statement:** All data generated or analyzed during this study are included in this article.

**Acknowledgments:** We thank anonymous reviewers for their comments towards improving this manuscript.

**Conflicts of Interest:** The authors declare no conflict of interest.

## Appendix A

The approximated expression of the first three order term in Equation (25) is given below, and they are all calculated by the first seven order truncated phase Equation (A1).

$$
\begin{aligned}
\Phi_{deramp\_7th}(f_\tau, f_\eta) = &-\frac{4\pi(f_0+f_\tau)}{c} \sum_{i=2}^{7} \frac{i-1}{i} C_i \left[ P\left(f_\eta + \frac{2\pi(f_0+f_\tau)R_1}{c}\right) \right]^i \\
&- 2\pi \left(f_\eta + \frac{2\pi(f_0+f_\tau)R_1}{c}\right) \sum_{i=2}^{7} C_i \left[ P\left(f_\eta + \frac{2\pi(f_0+f_\tau)R_1}{c}\right) \right]^{i-1} \\
&- \frac{\pi f_\tau^2}{K_r} - \frac{4\pi(f_0+f_\tau)R_0}{c} - \frac{\pi f_\eta^2}{K_{r\_rot}}
\end{aligned} \tag{A1}
$$

$$
D_0(f_\eta) = -\pi \left(2f_\eta + \frac{4R_1 f_0}{c}\right) * \sum_{i=2}^{7} (-1)^{i-1} C_i \left[\frac{cd(f_\eta)}{2f_0}\right]^{i-1} - \frac{4\pi f_0 \left(R_{ref} + \sum_{i=2}^{7} (-1)^i \frac{i-1}{i} C_i \left[\frac{cd(f_\eta)}{2f_0}\right]^i\right)}{c} - \frac{\pi f_\eta^2}{K_{r\_rot}} \tag{A2}
$$

where

$$
d(f_\eta) = \left(f_\eta + \frac{2R_1 f_0}{c}\right) \tag{A3}
$$

$$
\begin{aligned}
D_1(f_\eta) = \ \pi \Bigg[ &\begin{pmatrix} -1920C_7 R_1^7 + 2240C_6 R_1^6 - 2688C_5 R_1^5 + 3360C_4 R_1^4 \\ -4480C_3 R_1^3 + 6720C_2 R_1^2 - 13340R_{ref} \end{pmatrix} * f_0^7 \\
+ &\begin{pmatrix} 10080C_7 R_1^5 c^2 f_\eta^2 - 8400C_6 R_1^4 c^2 f_\eta^2 + 6720C_5 R_1^3 c^2 f_\eta^2 \\ -5040C_4 R_1^2 c^2 f_\eta^2 + 3360C_3 R_1 c^2 f_\eta^2 - 1680C_2 c^2 f_\eta^2 \end{pmatrix} * f_0^5 \\
+ &\begin{pmatrix} 16800C_7 R_1^4 c^3 f_\eta^3 - 11200C_6 R_1^3 c^3 f_\eta^3 + 6720C_5 R_1^2 c^3 f_\eta^3 \\ -3360C_4 R_1 c^3 f_\eta^3 + 1120C_3 c^3 f_\eta^3 \end{pmatrix} * f_0^4 \\
+ &\left(12600C_7 R_1^3 c^4 f_\eta^4 - 6300C_6 R_1^2 c^4 f_\eta^4 + 2520C_5 R_1 c^4 f_\eta^4 - 630C_4 c^4 f_\eta^4\right) * f_0^3 \\
+ &\left(5040C_7 R_1^2 c^5 f_\eta^5 - 1680C_6 R_1 c^5 f_\eta^5 + 336C_5 c^5 f_\eta^5\right) * f_0^2 \\
+ &\left(1050C_7 R_1 c^6 f_\eta^6 - 175C_6 c^6 f_\eta^6\right) * f_0 + 90C_7 c^7 f_\eta^7 \Big] / \left(3360c f_0^7\right)
\end{aligned} \tag{A4}
$$

$$
\begin{aligned}
D_2(f_\eta) = \ \pi \Bigg[ &\begin{pmatrix} -96C_7 K_r R_1^5 c f_\eta^2 + 80C_6 K_r R_1^4 c f_\eta^2 - 64C_5 K_r R_1^3 c f_\eta^2 \\ +48C_4 K_r R_1^2 c f_\eta^2 - 32C_3 K_r R_1 c f_\eta^2 + 16C_2 K_r c f_\eta^2 \end{pmatrix} * f_0^5 - 32 f_0^8 \\
+ &\begin{pmatrix} -240C_7 K_r R_1^4 c^2 f_\eta^3 + 160C_6 K_r R_1^3 c^2 f_\eta^3 - 96C_5 K_r R_1^2 c^2 f_\eta^3 \\ +48C_4 K_r R_1 c^2 f_\eta^3 - 16C_3 K_r c^2 f_\eta^3 \end{pmatrix} * f_0^4 \\
+ &\left(-240C_7 K_r R_1^3 c^3 f_\eta^4 + 120C_6 K_r R_1^2 c^3 f_\eta^4 - 48C_5 K_r R_1 c^3 f_\eta^4 + 12C_4 K_r c^3 f_\eta^4\right) * f_0^3 \\
+ &\left(-120C_7 K_r R_1^2 c^4 f_\eta^5 + 40C_6 K_r R_1 c^4 f_\eta^5 - 8C_5 K_r c^4 f_\eta^5\right) * f_0^2 \\
+ &\left(-30C_7 K_r R_1 c^5 f_\eta^6 + 5C_6 K_r c^5 f_\eta^6\right) * f_0 - 3C_7 K_r R_1 c^6 f_\eta^7 \Big] / \left(32K_r f_0^8\right)
\end{aligned} \tag{A5}
$$

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
