# Peer review of "A Novel Ultra−High Resolution Imaging Algorithm Based on the Accurate High−Order 2−D Spectrum for Space−Borne SAR"

_remotesensing, doi:10.3390/rs14092284_

Round 1

Reviewer 1 Report

The paper is well presented and introduced. Overall, math is extensively described which could be good for who has experience in the topic and a little hard for the others, but the general idea is that it is well described and nothing is foggy.

Results on simulations seem promising so far.

From the scientific point of view I've got nothing to argue, good job, let's see in the future the behaviour on real data.

The weak point of the article is the quality of the English. I've started to point out some propositions with grammatical mistakes, like in the introduction

"In the traditional space-borne SAR system, many classic imaging algorithms possess the great ability with the simplifying air-borne imaging geometry assumptions like range Doppler algorithm (RDA) [7] and the chirp scaling algorithm (CSA) [8]."

 or a little later

"Meanwhile, these methods regard the three- and higher-terms of the signal spectrum as the phase error and ignored."

Also on page 6: "The total Doppler bandwidth of ultra-high resolution space-borne SAR echo is always far larger than the Pulse Repetition Frequency (PRF) [28] which limited by the range ambiguity and data volume."

But then I realised that there is a remarkable quantity of mistakes, so, please, a general and complete revision of the English is extremely due.

Good job!

Reviewer 2 Report

This paper deals with the imaging problem for space-borne SAR system. The authors propose a frequency-domain imaging algorithm based on high-order 2-D spectrum. Overall, the paper is interesting and the results are promising. I have the following comments:

  1. Why do you propose a frequency domain algorithm based on chirp scaling? The time domain imaging algorithms[1,2] are much more accurate. Please emphasize the significance of the proposed approach in comparison with the time domain imaging algorithms. It’s also beneficial to compare the imaging performance of the proposed approach with the time-domain imaging algorithm.
  2. The spectrum model is established based on infinite polynomial range model. However, in (3), what’s the values of $Y_5$, $Y_6$, and so on? How can you get the values of R_i? In (15), what’s the values of $C_7$, $C_8$, and so on? How do you construct the deramp phase in (25) as it’s related to infinite terms? These questions are unclear to me. It’s better to illustrate the aspect clearer.
  3. Motion error is a key problem in the practical application of SAR image formulation[3]. For space-borne SAR systems, The imaging performance of LEO SAR is also affected by the motion errors. Is it possible for the proposed approach to be integrated with some autofocus algorithms? Adding some experiments is intractable, however, some discussion is beneficial.
  4. A language revision is necessary to fit some minor grammatical errors, e.g. the misuse of prepositions affects the readability of paper.
  5. The authors should pay attention to the format of references.

[1] G. Xu, S. Zhou, L. Yang, S. Deng, Y. Wang and M. Xing, "Efficient Fast Time-Domain Processing Framework for Airborne Bistatic SAR Continuous Imaging Integrated With Data-Driven Motion Compensation," in IEEE Transactions on Geoscience and Remote Sensing, vol. 60, pp. 1-15, 2022

[2] H. Xie et al., "Fast Factorized Backprojection Algorithm for One-Stationary Bistatic Spotlight Circular SAR Image Formation," in IEEE Journal of Selected Topics in Applied Earth Observations and Remote Sensing, vol. 10, no. 4, pp. 1494-1510, April 2017

[3] W. Pu, "SAE-Net: A Deep Neural Network for SAR Autofocus," in IEEE Transactions on Geoscience and Remote Sensing, doi: 10.1109/TGRS.2021.3139914.

Reviewer 3 Report

The manuscript provides a novel imaging algorithm for the ultra-high resolution space-borne SAR. A high precision range model derive from MESRM is introduced to the echo’s 2-D spectrum analyzing. 

The solution of the task to be solved is quite interesting. However even the manuscript is well organised and the introduction is supported with sufficient number of references there still some minor questions.

1. The manuscript will benefit if the authors provide a structured abstract, that covers the following aspects: the background (in which the authors should place the issue that the manuscript addresses in a broad context and highlight the purpose of the study), the methods used to solve the identified issue (that should be briefly described), a summary of the article's main findings followed by the main conclusions or interpretations. In the abstract the authors must also declare and briefly justify the novelty of their work. The authors should present in a clearer manner the above-mentioned aspects: the background, the methods, the main findings, the conclusions, as in the actual form of the manuscript, the abstract offers information related only to some of these aspects and even so, their delimitation is unclear.

2. Please provide in the Introduction before paper organisation description the statement of the main gap that the authors try to solve.

3. In the Section 2. "Infinite Polynomial Range Model and Signal Spectrum» please provide references to the firstly used expression for the range history vector. The please add references to used expressions  if they are derived from other manuscripts or state that the used expressions are generally accepted.  
4. At the conclusion please provide a short paragraph about how the gap (from the remark 2) was solved.
